# Dog Welfare, Well-Being and Behavior: Considerations for Selection, Evaluation and Suitability for Animal-Assisted Therapy

**DOI:** 10.3390/ani10112188

**Published:** 2020-11-23

**Authors:** Melissa Winkle, Amy Johnson, Daniel Mills

**Affiliations:** 1Center for Human Animal Interventions, Oakland University, Rochester, MI 48309, USA; 2School of Life Sciences, University of Lincoln, Lincoln, Lincs LN6 7DL, UK

**Keywords:** welfare, well-being, behavior, shelter, companion animals, dogs, training/positive reinforcement training, evaluation

## Abstract

**Simple Summary:**

Benefits for humans participating in animal-assisted therapy (AAT) have been long documented; however, welfare considerations for the animal counterparts are still quite non-specific, often relating to more general concerns associated with animal-assisted interventions (AAIs). Providers of AAT have a moral and ethical obligation to extend the “Do No Harm” tenet to the animals with whom they work. Companion animals do not ask or voluntarily sign up to be a part of a therapeutic team and their natural traits of love and sociability can easily be misinterpreted and exploited. This article reviews the current state of animal-assisted interventions; it highlights the lack of sufficient evaluation processes for dogs working with AAT professionals, as well as the risks associated with not protecting the dogs’ welfare. Finally, the authors make recommendations for determining the suitability of specific dogs in the clinical setting and ensuring that the population, environment, and context of the work is amenable to the dogs’ welfare and well-being.

**Abstract:**

Health care and human service providers may include dogs in formal intervention settings to positively impact human physical, cognitive and psychosocial domains. Dogs working within this context are asked to cope with a multitude of variables including settings, populations, activities, and schedules. In this article, the authors highlight how both the preparation and operation of dogs within animal-assisted therapy (AAT) differs from less structured animal-assisted activities (AAA) and more exclusive assistance animal work; the authors highlight the gaps in our knowledge in this regard, and propose an ethically sound framework for pragmatic solutions. This framework also emphasizes the need for good dog welfare to safeguard all participants. If dogs are not properly matched to a job or handler, they may be subjected to unnecessary stress, anxiety, and miscommunication that can lead to disinterest in the work, overt problematic behavioral or health outcomes, or general unsuitability. Such issues can have catastrophic outcomes for the AAT. The authors propose standards for best practices for selection, humane-based preparation and training, and ongoing evaluation to ensure the health, welfare and well-being of dogs working in AAT, which will have concomitant benefits for clients and the professionalism of the field.

## 1. Introduction

For decades, dogs have accompanied their guardians to work in therapeutic environments [1]. The discipline of mental health has provided the foundation for other diverse categories of animal-assisted interventions. In the 1930s, Dr. Sigmund Freud’s Chow Chow, Jofi, joined him in psychotherapy sessions [2,3], where Freud found that Jofi helped facilitate sessions by lying beside non-anxious patients and moving away from tense or stressed patients [2,3]. Freud claimed that his patients would often disclose more by speaking through Jofi. Similarly, in the 1960s, Dr. Boris Levinson’s dog, Jingles, was a frequent presence in sessions [4,5]. Levinson was also able to help his patients achieve therapeutic breakthroughs through their communication with Jingles. His documentation of his experiences in the book “Pet-Oriented Psychotherapy” later led to him being called the father of animal-assisted therapy (AAT) [2,4,6]. Since then, dogs have visited patients in hospitals [7,8] residents in nursing homes [9], and students in schools [10], and their popularity has expanded from mental health to include professional teams in physical [11], occupational [12,13,14], speech [15], recreational therapy [16] and other related disciplines [17].

Benefits for humans have considerable documentation within the literature, though far less attention has been given to the welfare of the dogs who work in these environments. Companion animals did not ask or voluntarily sign up to be a part of a therapeutic team and their natural traits of love and sociability can easily be misinterpreted and exploited. In Part 1, the authors discuss the current state of animal-assisted interventions and the lack of specific formal evaluation processes for dogs working in therapeutic settings, as opposed to other forms of animal-assisted intervention (AAI). In Part 2, the authors highlight the ethical obligation of professional health and well-being practitioners working with dogs and the potential risks associated with not safeguarding the welfare of the dogs participating in therapeutic sessions. Finally, the authors provide recommendations for determining the suitability and sustainability of a dog’s inclusion in AAT practice. These concepts are a culmination from organizational works, workshops, and presentations by the authors over recent years [18,19,20,21,22,23,24,25,26], and their roles in AAI course curriculum development [27]. While this paper focuses on dogs working in a therapy setting, the recommendations discussed in this paper may apply to other species who work with AAI providers in therapy settings.

## 2. Part 1: The Unique Character of Animal-Assisted Therapy within the Spectrum of Animal-Assisted Interventions

### 2.1. Animal-Assisted Interventions

The inclusion of dogs in AAIs appears increasingly in the literature for leisure visitation activities and as adjuncts to education, health care, and human service provisions. A Google Scholar search of “dog and AAI” returned fewer than 10 articles for 2009 and around 200 a decade later in 2019. Though the term AAI is often the catch-all term in the literature, the environments, populations, activities, credentials of the person delivering the service, expectations of the animals, theoretical approaches and scope of sessions vary greatly. This may complicate any prescriptive considerations for normative criteria for animal selection, preparation, evaluation, suitability and welfare in the context of the work that animals assist with.

The current terminology in AAIs adds to the confusion between leisure and professional representation of services and the scopes of practice to participants, which undermines rational theoretical applications, standards and competencies, as well as the ability to robustly study client outcomes or research specific interventions [28,29,30]. The lack of differentiation often leads to inaccurate generalizations, false expectations, and the potential for failure in the application and study of the impact of a given category of AAI [28,31]. For these reasons, the requirement for evidence-based practice, standards, competencies, and specialty credentialing within the professional subdisciplines is difficult to enforce. By identifying the expectations of the dogs in any specific type of activity, it should be possible to develop more appropriate guidelines for at least this aspect of the work.

AAI is an interdisciplinary umbrella term that encompasses the specific categories of animal-assisted activities (AAA), animal-assisted education (AAE) and animal-assisted therapy (AAT) to promote well-being and benefits for humans, and provide a positive experience for the animals without force, coercion or exploitation [30]. AAA, AAT and AAE services are all important, but differ in scope of provision. AAA service providers, on their own, do not typically require any professional credentials to offer visiting services. Professional AAE and AAT differ from AAA in that there is typically a prerequisite of human credentials; for example, a license or degree in a specific professional discipline such as health care, human service, or education. However, this is not always the case [32]. Both AAE and AAT providers should follow the same recommended professional processes for engaging with their clients or student, such as obtaining informed consent, offering formal evaluation, establishing short- and long-term goals, and measuring and documenting progress.

Other terminology that is confused with, but not directly related to AAIs, includes that which is associated with the growing number of Emotional Support Animals (ESA). ESAs are often pets and may require no special training, but provide emotional support and a sense of comfort for their guardians or others who have diagnosed psychological disorders. As with all pet dogs, ESAs can be therapeutic to their guardians, though they are not nor should they be considered formal therapy dogs [33].

Assistance animals are also a very different category of subjects compared to those involved with AAIs or ESAs. Assistance dog is a generic term for a guide, hearing, or service dog that is specifically trained to do tasks to mitigate the effects of an individual’s disability [34]. As assistance dogs are an extension of the human, dogs designated as service dogs often gain legal protections and more public access, particularly in the United States of America, from the Americans with Disabilities Act, Title II and Title III [35].

The scientific literature within AAIs has grown exponentially over the past decade; however, specific literature to support targeted interventions involved in AAT is minimal. There are few AAT specific processes for program development, dog selection, handler and dog team preparation and training, and team evaluation [36,37,38,39,40]. As noted above, the inconsistent terminology used in the literature makes it difficult to identify AAT-specific processes and literature. However, literature searches most often result in articles and research related to AAA, police/military working dog, or assistance dog traits or preparation processes and are frequently used to fill the void in AAT-specific content [41,42]. This, in and of itself, can be problematic because without an overarching, regulatory body to oversee policies and procedures for evaluation, most organizations develop their own [43]. It is therefore not surprising that a national 2020 study of nearly three dozen AAA evaluation and registration organizations found no consensus on several factors relating to the criterion for passing evaluation [43]. This included the length of time a dog lived with the handler prior to evaluation, the frequency of re-evaluation of the dog, vaccination requirements, amount of time and frequency of visits, restriction of activity if the dog showed signs of illness, or appropriate humane training methods [43].

The preparation of the dog for the work required in AAT needs to match the population, environment, and context of the practice. Some may argue that having any standard is better than operating with no standard at all. While there is some merit to this sentiment, using incomplete or incorrect standards can lead to poor dog welfare and exploitation.

It is fair to say that AAA, AAT and assistance dog work share similar foundational theories including those relating to the human–animal bond, human–animal interactions, biophilia, and biocentrism. However, the mechanisms of change and ways in which the teams work are very different. As we demonstrate below, much of the AAA and assistance dog selection, training and evaluation processes may actually be incompatible with expectations in AAT. Therefore, the processes used in AAA and assistance dog work may not be as appropriate for AAT as previously thought.

### 2.2. Differences between AAA and AAT

AAA human–dog teams visit public spaces in hospitals, facilities, and places where they may not have control over the environment or people. Many AAA organizations share similar skills and aptitude tests based on how the teams are expected to interact with participants [44,45,46] and perhaps based on their organizational standards of practice, which rightly have safety at their core for the nature of visitations [47,48]. Some even offer a certification. However, this does not come without problems. One concern is that some organizations offering certification have a financial or reputational interest in the success of the dog, which can result in a conflict for the best interest of the dog and the clients they work with [49]. Other concerns relate to organizations offering both the education and evaluation of the dog, which potentially results in a conflict of interest. Related protocols may also not have been assessed for their scientific validity [50]. Studies have described AAA organizational procedures for dogs who work in an AAT capacity [42,50], but whether these AAA procedures and evaluations accurately and consistently measure how a dog works contextually in AAT is not known as the nature of the work is so different. Professionals who offer AAT can look for direction from organizations that have been built by and for professionals who incorporate AAT services, and some discipline-specific organizations are in the early stages of including AAT guidelines.

Several professional organizations have made progress in publishing AAT-specific competencies while others have included AAT as a recognized practice area. For example, the American Counseling Association published the first set of professional competencies in 2016 [51]; the American Psychological Association Human Animal Interaction Section 13 Division 17 recently released a list of required competencies and ethical guidelines [18]; and the American Occupational Therapy Association (American Occupational Therapy Association, n.d.) recognize animal-assisted therapy as a specific practice area. There are many more commercial and non-profit organizations with some form of internal regulations, guidelines, and other valuable documents. What all of these competencies, ethics, standards, and guidelines have in common is a recognition of the need to protect the safety and welfare of the animals in practice (which includes using humane training methods) and the participants they work with. While professionals may have degree-level qualifications to serve the human participants, incorporating AAT also requires knowledge about dogs including their body language and behavior at the level of species, breed, and individual characteristics. There may be a risk of injury to the animal or client if the provider does not predict and respect the needs of the dogs. They need to take appropriate action to mitigate the causes of signs of discomfort and distress immediately.

In addition, it needs to be recognized that the dog could develop negative associations with the therapist, the client, the space, or develop chronic health problems [52]. When an AAI clinician honors and acts upon the messages shared by their dogs about the dogs’ wants and needs, their clients witness the compassion and care extended to the dogs to which clients can generalize to themselves. The safety that clients feel with the clinicians is often at the crux of effective treatment. Conversely, ignoring the needs of the dogs can send potentially harmful messages to clients about their own needs, wants and self-advocacy.

While human provider competencies are becoming more streamlined [51,53], there are no current, comprehensive normative evaluations for AAT due to the heterogeneous nature of intradisciplinary and interdisciplinary practices and subspecialties. Given this lack of AAT processes, AAA procedures and evaluations continue to be used, regardless of their incompatibility with how dogs function in AAT as part of the therapeutic process and goal achievement. Without sufficient evaluations, there is the potential of dogs being exploited, having negative experiences, and experiencing poor well-being [54,55,56].

Most current organizations set at least minimal requirements for dogs working in an AAA setting. These requirements stress the need for participant and dog safety. The dogs and handlers are evaluated against the minimal requirements which may attempt to mimic possible settings to which the dogs may be exposed. During these evaluations, dogs may be expected to accept, including but not limited to, the following:Remain on leash [45,57,58] with limited ability to roam freely [47,58];Avoid vocalizations such as barking [47]Work only under the direction of the handler [47]Be touched by unknown individuals on sensitive areas such as their feet or tails [48]Receive no food rewards during visits [58]Concede to social pressure of individuals or groups [48,58]Have brief interactions with people that are determined by someone other than the dog [47,48,58]Be evaluated and provide visits in novel environments with people they do not know [47,48,58]Be physically placed in a participant’s lap [47,58] potentially with the dog’s head controlled or forced in position by a handler at this time [47]

In some organizations, if a dog whines, barks or pulls away from an evaluation helper when the dog’s human guardian leaves the room for a minute, it results in an automatic failure of the evaluation. Similarly, demonstration of pulling, shyness, or resisting any part of the evaluation when the guardian is handling the dog would also mean automatic failure [58]. The initial education required by the handler prior to evaluation and registration varies according to organizations. Some organizations require an initial in-person evaluation of dog and handler with online renewals and no re-evaluation of the initial required skills [47,58], while other organizations require an in-person skills and role play-based initial evaluation and re-evaluations every two years [48].

Mongillo et al. [50] measured the difference between evaluated and registered AAI dogs and non-evaluated pet dogs in mock AAI scenarios. Dogs were subjected to unknown individuals who patted the tops of their heads, grabbed their harnesses, and hugged the dogs. They found that both the registered dogs and pet dogs showed similar signs of stress, which suggests that those dogs trained and evaluated for AAI were no better equipped to handle stressful AAIs than pet dogs [50,54].

By contrast, it has been the experience of the first author in AAT practices, that dogs often work off lead, greet participants (with some vocalizations welcomed), and work under the direction of the handler or the participant. Many professionals providing AAT have the luxury of focusing preparation and training on relationship-based techniques to elicit the human–animal bond, to gain the trust of their clients [59], and/or create relational moments [60]. Professionals are encouraged to advocate for their dogs by establishing ‘rules of engagement’, which include dog preferences for environment, population, activity, proxemics and touch, and to allow the dog to exit a situation as desired [54,55,61]. To decrease liability and improve animal welfare, professionals may be expected to complete a formal risk assessment and management plan, to have intermediate to advanced skills in AAT, knowledge of treatment planning and delivery techniques in AAI; as well as dog learning theory, training and welfare [51,61]. They should be qualified to screen clients for appropriateness to participate in AAT. This should consider, for example, evaluation of potential fears, allergies, health issues, differing cultural beliefs, history with animals and any potential link to domestic violence and other traumatic events [22]. Some AAT organizations recommend evaluation of the teams any time there is a change in population, environment, activity type, after prolonged periods away, and at least once a year [40,53]. The authors have not found any AAA organizations that evaluate for the specific situations in which dogs work, for example, working directly with children and people on the floor practicing exercises in an occupational therapy context. Furthermore, some AAA organization policies are very clear that the AAA-related insurance policies may not cover handlers and dogs who participate in animal-assisted therapy in paid working roles such as mental health providers who incorporate their dogs into practice [47,48,58].

### 2.3. Differences between the Assistance Dog Model and AAT

Assistance dogs are typically trained and placed to serve a single person and do so until they are ready to retire. The training and evaluation follow a prescriptive model with a specific set of tasks according to the type of placement they will fulfil. These include guiding, hearing and different types of service, although many organizations also provide some level of specialty training according to the recipient’s unique needs [34,62,63,64]. The dogs may be trained to ignore other people, to have high levels of obedience, and low impulsivity [62]. They may also be trained to know when not to follow the direction of their handlers as their lives may depend upon the dog performing specific tasks [65,66]. Assistance Dogs International member organizations are required to provide placement training education for recipients in which the skills of the dog and handler are evaluated [34,65,67]. The authors were not able to identify consistent standards for ongoing yearly evaluations.

In contrast, a dog that works in AAT may work with one or more handlers, and with many individual clients or groups of clients over their careers. Dogs may be expected to independently seek out and greet clients and even include barking or other vocalizations in the greeting. For example, clients who do not typically receive enthusiastic reactions from others might feel very special having a dog bark excitedly to them when they enter the room. The verbal and physical excitement may strengthen the initiation and facilitation of the therapeutic process. While they are expected to have manners and obedience, ideally the dogs should be able to choose whether or not they participate in the AAI session. This may be seen as an opportunity to use self-advocacy skills, empathy building or to use problem solving and non-verbal communication skills. The environments, client populations, activities and performance expectations of the dog vary as no two practitioners or sessions are going to be exactly alike. The variability of AAT preparation, training, intervention, and evaluation are significant, which calls for a more predictive analytic model. There are as many possible responses from dogs as there are differences in the way clinicians run their sessions.

## 3. Part 2: Professional Responsibility Concerning the Inclusion of Animals in Therapy

### 3.1. The Moral Imperative

Health care and human service professionals must consider their ethical responsibilities, including the changing culture towards the “use” (and possible exploitation) of animals, their welfare, and their well-being. Most professionals have core values that include boundaries of competence, altruism, and prudence (clinical and ethical reasoning skills) that help guide interventions with clients. Additionally, most codes of ethics include the core biomedical tenets of biomedical practice described by [68] of (a) non-maleficence (refraining from actions that cause harm, impair practice or compromise safe and competent services), (b) beneficence (doing good and preventing harm; safety and removing conditions that will cause harm), (c) autonomy (allowing control by the individual), and (d) justice (fairness), together with expectations of professional development for any specialty, complementary or alternative practice.

As professionals consider these responsibilities, the authors call attention to how they are implemented within the context of AAT, especially with respect to the dogs that work within this realm. It is generally accepted that the value of dogs extends beyond their instrumental value and they have the right to moral consideration as sentient beings [69,70]. The four core tenets of biomedical practice outlined above must extend to the dogs involved as well, which means doing right by the dog even when it does not serve the client.

In addition, health care and human service provision carries inherent liability risks and the addition of live animals naturally increases these risks. The requirements of professional development, continuing education, and competency prior to practicing in specialty areas, such as AAT, are meant to decrease the risks and protect clients and the dogs that work within AAT. Thus, the professional practice of AAT mandates competency in several dimensions of animal science including the health, behavior and welfare of the species with which an individual works that directly and indirectly impact on the competency and safety of the AAT. The authors suggest that a working knowledge and application of animal learning theory, interspecies communication and humane training techniques are certainly foundational skills that decrease some of the risk and are a clear requirement for sound ethical practice.

The shorter lifespan of the dog, compared to humans, remains a frequently overlooked area. Faster aging can result in a notable decline in the dog’s physical skills and abilities, cognitive and emotional processes, and general preferences of likes and dislikes. Furthermore, as dogs age and experience declining vision, hearing, and mobility, their tolerances and preferences change [71]. These changes are often associated with chronic conditions which may alter both performance and risk which need to be appreciated [72]. More frequent contextual evaluation can identify changes that would support the health, welfare and well-being of the dog and working lifespan as well. Nonetheless, AAT continues to lack formal processes such as annual registration evaluations that may be applicable to the wide range of practices including recognition of the dog’s need to retire and retirement transition protocols.

### 3.2. A Framework for Including a Dog in a Professional AAT Team

As already noted, the processes currently used in AAA and assistance dog models have marked differences from what is needed in AAT. There is also enormous variability between dogs and how they work with their handlers [49]. In order to create an effective working model for AAT, the authors propose a systematic process in line with the recommendation of [49] that considers matching a dog with the handler and the specific job characteristics, more individualized team preparation and training, and a more frequent contextual evaluation process that allows for the variables inherent in AAT practice. The attributes of ideal dogs to work in AAT vary with what one hopes to achieve with them. The therapy settings, therapy environments, participant groups, procedures, intended therapy outcomes, forms of interactions between humans and animals, and tasks the dog fulfils, all need to be carefully considered as part of this process, and no single dog is likely to be excellent or appropriate in all situations. Once these unique characteristics are identified, they can then be evaluated and re-evaluated over time. The authors elaborate on this within the context of an ongoing cycle, as illustrated in Figure 1.

#### 3.2.1. Handler Profile

In AAT, it is common for professionals to own and handle dogs that they work with, and it is important to ensure that they are a good match. Relationship development and foundational dog considerations (welfare and well-being practices) begin in their living environment, where a dog will spend much of its time. By creating a specific handler profile in relation to the AAT, it becomes clearer as to what type of dog would best match the handler’s lifestyle which might include activity level, personality, leisure activity interests, dog training and skill level, etc. This profile creates an ideal opportunity for ensuring compatibility between personal (domestic) needs and the working environment including program development needs. The program development needs include, but are not limited to, risk assessment and management, evaluation of clients for appropriateness for AAT participation, continuing education, commitment requirements, and ability to manage clients, equipment, intervention plans, while also tending to the dogs’ needs, etc.

#### 3.2.2. Job Characteristics

Selection of the right dog for the job includes knowing information such as, but not limited to:(A)Breed restrictions(B)Specific client population demographics (physical and cognitive implications as well as client suitability)(C)Client population treatment categories (physical disabilities and potential for unnatural human postures and specialized equipment; cognitive status to comprehend appropriate interaction with the dog)(D)Psychiatric related issues that require medications that may alter a human’s state (humans becoming volatile in emotional transactions)(E)Environmental setting (small clinic, large hospital, indoors/outdoors, individual room/large open space, etc.)(F)Sensory sensitivities or distractions to which the dog may be exposed (visual, auditory, olfactory, tactile, etc.)(G)How the dog is to be part of the environment and sessions (on/off leash, physically sedentary or active, more engaging or more responsive, natural versus trained behaviors, able to remove itself or communicate when it is not interested in working, etc.)(H)Identification of work frequency and duration, number of handlers and clients in a typical day

Additionally, the dog must have a place, such as a bed or crate, away from humans and activity to rest. Dogs generally do well with routines. Participation expectations regarding the types of activities in which dogs will be participating (e.g., talk therapy, dog training, physical activities between dogs and humans) also need to be clearly articulated. In some situations, there may be more than one dog available to participate in sessions. In these settings, each dog would require an individual profile with the dog’s needs and preferences clearly identified to determine which dog would be best for each interaction. Each therapy session should also have a session plan [73] with desired outcomes of that session that factor in the potentially different responses to the dog by the client and by the dog to the client where each specific intervention proposed to evaluate plausibility.

#### 3.2.3. Animal Profile

There is a “general” set of skills and capacities that any dog who participates in therapeutic settings must have (including traits which should not be present), and while a discussion of these is beyond the scope of the current article they have been considered elsewhere [49]. In short these include a robust temperament, adaptability/flexibility, adequate training status and responsiveness, secure attachment with the handler, self-motivation for the job, quick recovery when startled, and willingness to engage in the sessions (See Table in [49] for further details). The authors focus here on several other significant considerations to keep in mind when matching a dog to a potential job. Knowing the animal’s history can help to identify any possible behavior issues, fear responses and recovery time to novel people, places and things in the environment. For example, the degree of anxiety or reactivity to specific situations may exclude a dog from working entirely, or it may just require a contingency plan should the rare situation occur. A history of protectiveness towards people, places or things can impact if, how and with whom a dog works, as this may be a liability. In some instances, a solid management plan may be all that is needed if the event is rather unlikely in a particular AAT setting. Understanding breed-typical traits and individual dog preferences can help shape the types of people and activities that would keep the dog engaged and in anticipation for the work, while also influencing the time of day, frequency and duration of sessions. However, it must be appreciated that breed-typical traits are just the norm and that enormous variability exists around this; they are not therefore a reliable predictor of differences between individuals [74].

Imagine a dog who does not actually want to be petted nor sit in a designated place for long periods of time, being asked to assist in talk therapy on a couch being stroked for hours during the day. Conversely, consider the same dog working in a physical therapy unit where participants are standing on a balance board throwing the ball for the dog for several repetitions, then doing an agility course to facilitate the participant working on ambulation and weight shifting. The latter job may be ideal for this specific dog and the dog will likely demonstrate signs of enjoyment in its tasks. Other factors for consideration are dogs who enjoy learning and training tasks and dogs who demonstrate consistency in their responses such as often communicating enjoyment or repeatedly removing themselves if they are uncomfortable given situations.

A dog will typically need to possess a sociability towards unfamiliar individuals and a curiosity towards various activities. The dog should also demonstrate patience towards participants who fail to complete an activity that would normally bring gratification to the dog. Proxemics are also a consideration. Similar to people, dogs are believed to have personal space preferences, and it may be useful to note differences in behavior and signaling according to whether the dog moves into a person’s space, or if a person moves into the dog’s space. Another powerful tool in AAT is the way the dog demonstrates interest in people, and the ability to work directly with the participant rather than only working for the handler.

Every dog working in AAT should have the right of access to solid health, welfare and well-being practices. By ensuring the dog’s profile complements that of the handler and the majority of the job description, large strides can be made towards achieving this. If practitioners are doing AAT correctly, the dog will display signs of enjoyment and want to return to sessions to work with participants again and again.

#### 3.2.4. Team Preparation and Training

The training that the team will require is largely based on strategies (goals), tactics (intervention plans to achieve goals), and logistics (coordination of what is required of all parties during the session). However, AAT human–animal team preparation has several layers, the most important of which is a strong positive relationship between the dog and the handler. The handler needs to be knowledgeable in dog communication and learning theory so they can accurately, and without personal bias, identify the dog’s comfort level and use the best training approach for a given situation. Handlers should use humane, positive reinforcement training techniques that do not involve force or coercion as this will build a dog’s confidence for the complex nature of AAT. The handler is seen as a secure base for the dog from which to operate [75] and using training techniques that harm or frighten the dog can damage that secure base. Allowing dogs to have autonomy and choice in the training and practices of AAT will likely improve the dogs’ engagement as they will learn on their own terms how to overcome the things they may not be sure of, or empower the dog to leave a situation entirely. Noting if dogs can quickly recover from a situation when comforted by their handlers also offers information about coping skills in novel situations. The handler should be clear and consistent with the ability to advocate for the dog when necessary. Knowledge of typical puppy development, fear periods, and socialization for people, places, things, and sensory experiences can impact how a dog may likely engage in the future. Handlers should also recognize that a dog’s preferences can change over time; things the dog enjoyed at 2 years old may be very different at 7 years old. Participation in a variety of enrichment and relaxation protocols can improve a dog’s quality of life and coping skills when the need arises. Not every client will want to participate in or be an appropriate fit for AAT; accordingly, dogs should be comfortable being left alone under different circumstances and at short notice.

Although some standards and evaluations of AAA organizations are incompatible with AAT expectations and allowances; being evaluated for and participating in volunteer visiting can be a useful way for someone to get to know their dog. It may also give the dog-handler team the opportunity to engage in an activity that involves novel environments and populations in a more managed situation. In addition, it gives the team experience with evaluation protocols and some solid interaction opportunities without the added responsibility of balancing a participant’s goals, a treatment plan, and equipment.

In an ideal situation, as the time comes closer to beginning AAT, time spent in the actual working environment, after hours, can afford dogs the opportunity to explore where they will work, rest and relieve themselves. It allows the dog to process environmental stimuli and to habituate to certain things in the environment. Introductions and initial interactions to the staff and in-services can be offered during these times so that positive associations may be made. AAT session simulations are a great way to assist with generalization of previously learned skills or the default behaviors required in a given context (e.g., if a dog sees a yoga mat being unrolled, this may signal that it should go and lay at the end of it, wherever this action occurs). Setting up routines for work, rest, enrichment and a safe space to implement these requirements carry a lot of value in employing a dog to work in AAT. By giving the dog these opportunities, it will increase the dog’s sense of safety and security so they know what to expect and recover quickly should the unexpected happen [76].

#### 3.2.5. Evaluation and Re-Evaluation

The heterogeneous nature of AAT requires a predictive model to evaluate the processes and skills of the teams working in such diverse approaches. While daily evaluation (observation) of the dog is done by the handler to identify behavior, level of interest and participation, at least annually, an objective, third party evaluation should take place in the environment, with a representative client population (age, disability, etc.), and with the types of activities they are expected to participate in (therapy equipment, floor time with participants, etc.). Re-evaluation should occur any time there is a change in population, environment, activity type, after prolonged periods out of sessions, and at least once a year [40,53]. The conditions under which the dog is expected to participate such as on/off leash, working from a distance, working in distractible settings, working with individuals or groups also provide a greater opportunity to evaluate not only performance, but also to construct remediation plans for areas in which performance if found lacking. It can be useful to break the evaluation down into skills, behavior and obedience. The evaluator should have training and experience in interpreting behavior and administering any specific tests that are used. Unfortunately, the quality of most rating and behavior tests is either unknown or poor in terms of both validity and even reliability [41] and this is an area which requires urgent scientific attention.

However, validated behavior profile instruments such as the Canine Behavioral Assessment and Research Questionnaire (C-BARQ) [77] and psychometric instruments such as the Positive and Negative Activation Scale [78], which measure sensitivity to rewards and aversive methods; the Dog Impulsivity Assessment Scale [79]; and the Canine Frustration Questionnaire [80] can be used on a regular basis to track changes and provide a useful record for monitoring purposes. Evaluators can also make systematic observations of the dog’s behavior, signals (lip licking, yawning, etc.) and cues to arousal (pupil dilation, panting, etc.) in the clinical context in order to make systematic evaluations of the dog’s emotional state [81]. Many dog trainers utilize consent tests to identify dog preferences, especially for touch and being hugged. However, if these tests are administered by the dog’s handler, the results should not be generalized to AAT participants. It is not recommended that participants take part in consent tests as there are safety issues for both dogs and participants. Unfamiliar handlers who are competent, however, may be used.

## 4. Conclusions

As the field of AAI has grown, so has the need for specific guidelines and standards for the welfare and well-being of the animals involved in these interventions. It is essential that these not only differentiate the demands placed upon dogs and other animals according to the type of AAI involved (e.g., AAA versus Service dogs versus AAT), but also the specific demands within these. While AAT is a structured activity with specific goals, it can also be highly varied with bespoke programs catering to the needs of the many different clients with which a single animal may work. There is therefore a need for professionals to recognize and adopt both general principles to safeguard the well-being of the animals with whom they work as well as bespoke action plans for each client, that include consideration of the expectations and demands being placed on the animal involved. Professionals have a further responsibility to objectively evaluate if the animal(s) they have available are suitable for these tasks and recognize that therapy should often proceed without an animal. Professionals must also recognize the limitations of many procedures used to evaluate animal behavior and welfare for their work and take measures to mitigate against these limitations in a responsible way, rather than underestimate the importance of formal assessments.

Health and human service professionals uphold an oath to “Do No Harm” (and beyond that, doing right by the dog) and that tenet must also extend to the dogs who are working in a professional capacity with the clinician. Selecting a dog to work in this setting without appropriate training, suitability evaluation, and good knowledge of dog body language, can result in the dogs’ welfare and well-being being jeopardized. The authors recommend a cycle of initial evaluation then annually evaluating or re-evaluating to ensure positive health and welfare for the dogs. This model includes having the professional develop a handler profile, a list of job characteristics specific to their work environment and needs, the dogs’ profiles that identify what will be realistically required of the dogs along with what traits would be most amenable, a plan for team preparation and training, and then evaluating or re-evaluating the dog—ideally on an annual basis.

## Figures and Tables

**Figure 1 animals-10-02188-f001:**
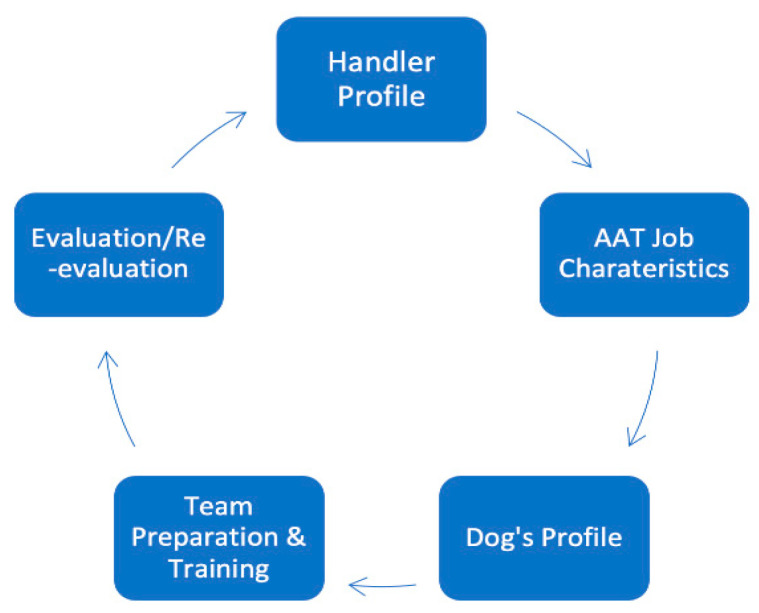
Framework for selection, preparation and evaluation of dogs working in AAT.

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
