# Peer review of "Dog Welfare, Well-Being and Behavior: Considerations for Selection, Evaluation and Suitability for Animal-Assisted Therapy"

_animals, 2020, doi:10.3390/ani10112188_

Round 1

Reviewer 1 Report

This manuscript addresses a very thorny issue in AAI and AAT; namely, the welfare of the dogs as they participate in these activities. It makes very important points concerning policies and practices. For example, lines 122-127 describe the lack of consensus among various groups that are responsible for registering or certifying therapy dogs. Another important point is the role conflict when the person who has worked with a handler/dog team in class then becomes the evaluator (lines 148-149). There may, in this case, be a vested interest in a successful outcome. Although this article isn't about training therapy dogs, it might also mention that harsh approaches are incompatible with preparation for therapy dog work.

One point that bears mentioning is that the current situation tends to place faith in the handler (or guardian, to use the authors' terminology) about deciding which activities are a good match for the dogs' strengths. People can become so determined to volunteer with their dogs that their enthusiasm overshadows doubts about the suitability of the specific situation for the specific animal. To illustrate, a very dedicated therapy dog handler brought a dog to an event too soon after it had surgery for the removal of several benign tumors, in my opinion. The dog wore a t-shirt to that he would not be touched there, but he is 9 years old and probably needs to be retired, rather than pushed to participate. A "just trust the handler" approach has its limitations.

Two more recent developments with Alliance of Therapy Dogs may bear mentioning. First of all, it is possible to purchase additional insurance to cover the team when they are at the handler's place of work rather than volunteering. Line 230 states otherwise. Secondly, due to Covid-19, the rules for assessment have been modified/relaxed. Whereas there was a handling test and three observations on three different days and places, (two of which had to be medical facilities where dogs would encounter orthopedic equipment), the observations now can be conducted at any dog-friendly location (e.g., park, store).

Given the tenor of this piece, a "do no harm" motto may not go far enough. It may be more the case of "doing right by the dog". At times, people do not see the signs and they need a more objective look at their dog's behavior. For instance, one therapy dog group made a video of their dogs visiting a school and, when one person watched her dog (who had since died from bone cancer) walking rather gingerly down the hallway, her eyes filled with tears. She later said, "I should have realized she wasn't herself."

Minor edits (mainly with possessives):

Line 18 should be dogs' welfare

Line 32 would read better as: unsuitability. Such issues can have...

Line 45 omit period before parenthesis

Line 73 "less than 10" should be "fewer than 10" (for quantities that can be counted, fewer is the correct word

Line 195 should be dog's head

Line 197 should be dogs' human guardian

Line 271, insert comma, should be:  , outlined above,

Line 315 should be dogs' needs

The issues highlighted in this manuscript and the current literature review make it a must read for those involved in therapy dog work. It raises awareness about the sometimes taken-for-granted assumptions that could compromise the bond of trust between the human guardian and the therapy dog.

Author Response

Thank you for your very thorough review of our commentary. We have revised the article according to your recommendations. I have added the notes/revisions below. 

This manuscript addresses a very thorny issue in AAI and AAT; namely, the welfare of the dogs as they participate in these activities. It makes very important points concerning policies and practices. For example, lines 122-127 describe the lack of consensus among various groups that are responsible for registering or certifying therapy dogs. Another important point is the role conflict when the person who has worked with a handler/dog team in class then becomes the evaluator (lines 148-149). There may, in this case, be a vested interest in a successful outcome. Although this article isn't about training therapy dogs, it might also mention that harsh approaches are incompatible with preparation for therapy dog work. Yes, we agree. This is a very important point and we did add a paragraph under the Moral Imperative section.

One point that bears mentioning is that the current situation tends to place faith in the handler (or guardian, to use the authors' terminology) about deciding which activities are a good match for the dogs' strengths. People can become so determined to volunteer with their dogs that their enthusiasm overshadows doubts about the suitability of the specific situation for the specific animal. To illustrate, a very dedicated therapy dog handler brought a dog to an event too soon after it had surgery for the removal of several benign tumors, in my opinion. The dog wore a t-shirt to that he would not be touched there, but he is 9 years old and probably needs to be retired, rather than pushed to participate. A "just trust the handler" approach has its limitations. Yes, we have seen this too and could not agree more. We added an additional sentence referencing this in the paragraph about the dog's life span.

Two more recent developments with Alliance of Therapy Dogs may bear mentioning. First of all, it is possible to purchase additional insurance to cover the team when they are at the handler's place of work rather than volunteering. Line 230 states otherwise. Secondly, due to Covid-19, the rules for assessment have been modified/relaxed. Whereas there was a handling test and three observations on three different days and places, (two of which had to be medical facilities where dogs would encounter orthopedic equipment), the observations now can be conducted at any dog-friendly location (e.g., park, store).

Given the tenor of this piece, a "do no harm" motto may not go far enough. It may be more the case of "doing right by the dog". At times, people do not see the signs and they need a more objective look at their dog's behavior. For instance, one therapy dog group made a video of their dogs visiting a school and, when one person watched her dog (who had since died from bone cancer) walking rather gingerly down the hallway, her eyes filled with tears. She later said, "I should have realized she wasn't herself." Yes, we concur and included that sentiment.

Minor edits (mainly with possessives): All grammatical edits were made.

Line 18 should be dogs' welfare

Line 32 would read better as: unsuitability. Such issues can have...

Line 45 omit period before parenthesis

Line 73 "less than 10" should be "fewer than 10" (for quantities that can be counted, fewer is the correct word

Line 195 should be dog's head

Line 197 should be dogs' human guardian

Line 271, insert comma, should be: , outlined above,

Line 315 should be dogs' needs

The issues highlighted in this manuscript and the current literature review make it a must read for those involved in therapy dog work. It raises awareness about the sometimes taken-for-granted assumptions that could compromise the bond of trust between the human guardian and the therapy dog. Thank you very much.

Reviewer 2 Report

Hi,

I have attached a PDF with the edits in the paper as comments. But if the authors can't see them, they are all reproduced below.

This paper sets out to describe the current state of some forms of therapies and assistance by dogs for people with mental health disorders and/or physical impairment. It is a timely paper that suggests a starting place for AAT, specifically, to develop guidelines for selecting, training and working with dogs in clinical settings while preserving the dog's welfare and health.

Simple Summary:
this is good but one suggested change:

Line 15: Remove "unconditional love". Do we really know if dogs love? And if they do, if their love is unconditional?
I would suggest removing this as it detracts from the science of this important discussion of the welfare of the dogs selected to do this work

Abstract:

Succinct and describes the paper.

Line 32: Should AAA be included here too? 

Keywords:
No changes.

Introduction:

Line 57: Same comment as before. Take out "unconditional love". Acceptance may be a better term.

Line 67: Do you mean other species doing the similar work in therapy settings? It is a little unclear and, as written, seems unfinished and odd.

Line 97-98: This sentence is a little unclear. Do you mean for them to do the same thing as each other or of someone else. Reword to clarify. eg "Both AAE and AAT providers should follow professional processes for engaging with their clients or students, for example, acquiring informed consent..."

Lines 100-105: References for the assertions in this paragraph are needed. 

Line 112-113: This introductory sentence is unclear- do you mean for practitioners to use to develop their own programs or implement someone else's therapies programs or studies of AAT in general? I am a bit confused as to the purpose of the AAT studies that are sparse.

Line 112-127: Just a question and comment- what is the situation for AAT? has any research looked at the same things for AAT? If not, I think this is interesting and a statement about this should be included.

Line 128: Clarity. Preparation of the dog for what??? AAA, AAT, either, both?

Line 134-137: "Much of the AAA and assistance dog selection, training and evaluation processes may actually be considered incompatible with expectations in AAT, therefore, the processes used in AAA and assistance dog work may not be as appropriate for AAT as might be thought." 'Sign posting' that this will be discussed next would be good. But some references would also help in here to support this assertion.

Line 161: Johnson et al., 2020); missing from reference list unless it should be Johnson et al n.d.

Line 170: I'd suggest discomfort and distress. I'd like to think the handler would be responding to the dog's discomfort before it becomes distressed.

Line 184-186: "Most current evaluation organizations set at least minimal requirements for dogs working in an AAA setting. These evaluative processes reflect the registration organizations' need for patient (and dog) safety and may attempt to mimic possible settings to which the dogs may be exposed." These two sentences are not clear. I'd suggest editing to something like: "Most current organizations set at least minimal requirements for dogs working in an AAA setting which stress the need for patient and dog safety. The dogs and handlers are evaluated against the minimal requirements which may attempt to mimic possible settings to which the dogs may be exposed. 

Line 197: dogs should be dog's

Line 202 and 203: in person should be in-person

Line 222: "..participate in AAT including fears, allergies, ..." This is a little clumsy. Can I suggest "..participate in AAT, considering, for example, fears, allergies, ...

Line226: This is also a bit clumsy and not clear.
"...specific situations in which dogs work including with children and people on the floor."
Can I suggest: "...specific situations in which dogs work, for example, working with children and people on the floor."

Line 268: "As professionals consider these responsibilities we call attention to...."This first part is not clear. Is there a comma or words missing? 

Line 269: "It is generally accepted that the value of dogs extends beyond their instrumental value and they have the right to moral consideration as a sentient being." Reference for this assertion?

Line 274: "...that risk." Should be '...these risks."

Line 276:"...protect our clients and the dogs that work with us. " this is not in the same 'voice' as the rest of the paper. change out the 'our' and 'us' for consistency.

Line 379: Remove extra space between "whether" and "the"

Line 382: New paragraph starting at "Every dog..."

Line 389-390: Missing a word or two... "However, AAT human-animal team preparation has several layers, the most important of which is a strong positive relationship with the handler."
Suggestion: "However, AAT human-animal team preparation has several layers, the most important of which is a strong positive relationship between the dog and the handler."

Line392: Handler’s should be Handlers

Line 393: Insert: positive reinforcement focussed training techniques
Focused is misspelt

Line 440: Brady et al. 2017 should this be Brady et al 2018 or is is a separate reference. Missing from reference list.

Line 537: Granger and Kogan 2000 Reference not in paper

Author Response

Thank you for the time and effort you put into a very thorough review. We appreciate your comments in helping improve this manuscript. We have reviewed each comment and made revisions as requested. I have added responses below. 

I have attached a PDF with the edits in the paper as comments. But if the authors can't see them, they are all reproduced below.

This paper sets out to describe the current state of some forms of therapies and assistance by dogs for people with mental health disorders and/or physical impairment. It is a timely paper that suggests a starting place for AAT, specifically, to develop guidelines for selecting, training and working with dogs in clinical settings while preserving the dog's welfare and health.

Simple Summary:
this is good but one suggested change:

Line 15: Remove "unconditional love". Do we really know if dogs love? And if they do, if their love is unconditional?
I would suggest removing this as it detracts from the science of this important discussion of the welfare of the dogs selected to do this work Changed

Abstract:

Succinct and describes the paper.

Line 32: Should AAA be included here too? 

Keywords:
No changes.

Introduction:

Line 57: Same comment as before. Take out "unconditional love". Acceptance may be a better term. Changed

Line 67: Do you mean other species doing the similar work in therapy settings? It is a little unclear and, as written, seems unfinished and odd.

Line 97-98: This sentence is a little unclear. Do you mean for them to do the same thing as each other or of someone else. Reword to clarify. eg "Both AAE and AAT providers should follow professional processes for engaging with their clients or students, for example, acquiring informed consent..." This sentence has been revised

Lines 100-105: References for the assertions in this paragraph are needed. References added in text and to the reference section

Line 112-113: This introductory sentence is unclear- do you mean for practitioners to use to develop their own programs or implement someone else's therapies programs or studies of AAT in general? I am a bit confused as to the purpose of the AAT studies that are sparse. This sentence has been revised

Line 112-127: Just a question and comment- what is the situation for AAT? has any research looked at the same things for AAT? If not, I think this is interesting and a statement about this should be included. This sentence has been revised

Line 128: Clarity. Preparation of the dog for what??? AAA, AAT, either, both? Revised

Line 134-137: "Much of the AAA and assistance dog selection, training and evaluation processes may actually be considered incompatible with expectations in AAT, therefore, the processes used in AAA and assistance dog work may not be as appropriate for AAT as might be thought." 'Sign posting' that this will be discussed next would be good. But some references would also help in here to support this assertion. References added

Line 161: Johnson et al., 2020); missing from reference list unless it should be Johnson et al n.d. Revised

Line 170: I'd suggest discomfort and distress. I'd like to think the handler would be responding to the dog's discomfort before it becomes distressed. Revised

Line 184-186: "Most current evaluation organizations set at least minimal requirements for dogs working in an AAA setting. These evaluative processes reflect the registration organizations' need for patient (and dog) safety and may attempt to mimic possible settings to which the dogs may be exposed." These two sentences are not clear. I'd suggest editing to something like: "Most current organizations set at least minimal requirements for dogs working in an AAA setting which stress the need for patient and dog safety. The dogs and handlers are evaluated against the minimal requirements which may attempt to mimic possible settings to which the dogs may be exposed. Revised

Line 197: dogs should be dog's Revised

Line 202 and 203: in person should be in-person Revised

Line 222: "..participate in AAT including fears, allergies, ..." This is a little clumsy. Can I suggest "..participate in AAT, considering, for example, fears, allergies, ...Revised

Line226: This is also a bit clumsy and not clear.
"...specific situations in which dogs work including with children and people on the floor."
Can I suggest: "...specific situations in which dogs work, for example, working with children and people on the floor." Revised

Line 268: "As professionals consider these responsibilities we call attention to...."This first part is not clear. Is there a comma or words missing?  Revised

Line 269: "It is generally accepted that the value of dogs extends beyond their instrumental value and they have the right to moral consideration as a sentient being." Reference for this assertion?  Added

Line 274: "...that risk." Should be '...these risks." Revised

Line 276:"...protect our clients and the dogs that work with us. " this is not in the same 'voice' as the rest of the paper. change out the 'our' and 'us' for consistency. Revised to reflect third person throughout paper

Line 379: Remove extra space between "whether" and "the" Revised

Line 382: New paragraph starting at "Every dog..." Revised

Line 389-390: Missing a word or two... "However, AAT human-animal team preparation has several layers, the most important of which is a strong positive relationship with the handler."
Suggestion: "However, AAT human-animal team preparation has several layers, the most important of which is a strong positive relationship between the dog and the handler." Revised

Line392: Handler’s should be Handlers Revised

Line 393: Insert: positive reinforcement focussed training techniques
Focused is misspelt Revised

Line 440: Brady et al. 2017 should this be Brady et al 2018 or is is a separate reference. Missing from reference list.Revised

Line 537: Granger and Kogan 2000 Reference not in paper Deleted

Reviewer 3 Report

General comments

The authors evaluate the current state of animal-assisted interventions and address the lack of specific formal evaluation processes for dogs working in therapeutic settings. They also highlight the potential risks associated with not safeguarding the welfare of these working dogs and provide a useful framework for including dogs in professional AAT teams.

While the essence of this paper is sound and contains a wealth of information, I feel that it is let down by its style of writing. The text consists of numerous statements or declarations that are followed by long lists of examples in parenthesis. Along with poorly constructed sentences, this often makes for tedious and frustrating reading. While I acknowledge that the authors are clearly experts in this field, I feel that this paper would benefit by being tightened up and made a bit more interesting.  Perhaps some of the long lists that are provided in parenthesis could instead be put in tabular form.

Sentences are often far too long and need breaking up. The authors sometimes use the first person e.g. ‘we’ and other times not. While the use of the first person is acceptable in an article of this nature (a commentary), it is important that the authors choose a style and then remain consistent throughout.

Please check on Animals’ rules regarding the use of American English versus British English e.g. organization vs. organisation; specialty vs. speciality; practicing vs. practising; behavior vs. behaviour.

Please also check on the rules of in-text referencing as many of the references are cited with no date.

To help improve this manuscript, I have made more specific suggestions below:

Title:

Companion Animal Welfare, Well-being and Behavior: Considerations for Selection, Evaluation and Suitability for Animal Assisted Therapy

This paper is about dogs participating in animal assisted interventions, specifically animal assisted therapy. The title should reflect this more fully. The term ‘Companion animal welfare’ is too broad.

Simple summary

Line 18: change to: ‘…the dogs’ welfare.’

Abstract

Line 29: By safeguarding ‘patients’, you are obviously referring to the humans participating in animal assisted therapy. In the veterinary domain, it is usual for the dog to be referred to as the patient. The way the sentence is worded, it sounds like the dogs are at risk of becoming patients if not properly matched to a job or handler.

Line 31: Couldn’t ‘disinterest in the work’ be construed as a behavioural outcome?

Line 34: ‘animal assisted therapy’ can be written as AAT here

Introduction

Lines 41-53: This opening paragraph of the Introduction relies very heavily on Ernst (2014). Are there other sources of literature on the history of dog use in therapy. Is there any prior literature on the history of animal-assisted interventions in the discipline of physical disability?

Line 45: The statement: ‘…would move away from the patient if he were tense.’ Should that not be:  he or she? Perhaps say: ‘…by lying beside non-anxious patients yet would move away from patients if they were tense.’

Line 50: Change to: Animal Assisted Therapy (AAT) as this is the first time that the term has been used in the Introduction.

Lines 50-53: A few references would be good here (and would counter the overrepresentation of Ernst, 2014 in this paragraph).

Line 63: Change to: AAT

Line 67: Change to: ‘…the constructs may apply to other species involved in AAIs.’

Animal-Assisted Interventions

Line 72: Need a reference or two after this first sentence

Lines 74-78: This is an excruciatingly long sentence. Please break it into two separate sentences:

e.g.:  Though the term AAI is often the catch-all term in the literature, the environments, populations, activities, credentials of the person delivering the service, expectations of the animals, theoretical approaches and scope of sessions vary. This may complicate any prescriptive considerations for normative criteria for animal selection, preparation, evaluation, suitability and welfare in the context of the work that animals assist with.

Line 81: Change to: ‘theoretical applications’

Lines 92 and 97 and 101: I am old fashioned and would not start a sentence with an abbreviation. I guess this is the Editor’s call.

Line 110: Change to: ‘…the United States of America…’.

Line 106-111: I would have regarded ‘Assistance animals’ or ‘service dogs’ as part of AAI. You do mention earlier that AAI is an umbrella term. I guess your point is that dogs specifically involved in tasks that mitigate the effects of an individual’s disability enjoy more legal protection than dogs involved in other forms of AAI (in the USA).

Line 119: Change to: AAT-specific

Lines 121-127: Another excruciating sentence! Please break it up.

Line 123: Change to ‘…found there was no consensus…’

Line 47: I have never heard the term: ‘conflict for the best interest’ but it seems to work.

Lines 154-156: How can we be certain that the AAT guidelines have been subject to ‘independent objective verification’ and have been assessed for their scientific validity and, therefore, provide an accurate and consistent measure of how a dog works contextually in AAT? I imagine that some AAT organisations may also have a financial or reputational interest in the success of the dog, which can result in a ‘conflict for the best interest’! As you assert later: ‘there are no current, comprehensive normative evaluations for AAT due to the heterogeneous nature of intradisciplinary and interdisciplinary practices and subspecialties’.

Lines 169-172: Please break up this laborious sentence.

Lines 172-175: Reference(s)?

Lines 186-196: This a ten-line sentence! I think it needs to be broken up. Maybe bullet points or at the very least the use of semi colons rather than commas between each ‘expectation’.

Line 193: Insert a comma (or semi colon) between these two expectations.

Line 199: Change to: ‘…any part of the evaluation when the guardian is handling…’

Lines 202-203: Change to: ‘…dog and handler with online renewals and no re-evaluation of the initial skills required skills (ATD, 2016b; TDI, 2018), while other organizations require an in-person skills …’

Lines 217: Change to: ‘…activity, proxemics and touch, and to allow…’

Line 220: Change to: ‘… to have intermediate to advanced skills…’

Lines 218-221: This is a confusing sentence. Should it be: ‘…to have intermediate to advanced skills in AAT treatment planning and delivery and dog learning theory, training and welfare…’

or:

‘…to have intermediate to advanced skills in AAT treatment planning and delivery, dog learning theory, dog training and dog welfare…’ ??

This confusion happens fairly frequently throughout this paper. Please carefully check your placement of commas and semi colons.

Line 222-224: This is another badly phrased sentence. Perhaps something like: ‘They should be qualified to screen clients for appropriateness to participate in AAT including having the ability to determine the clients’ potential fears, allergies, health issues, cultural beliefs, history with animals and exposure to domestic violence or other traumatic events (Winkle, 2015).’

Lines 224-226: This sentence also needs tightening up.

Line 227: What does ‘people on the floor’ mean?? Unconscious people? Inebriated people? People with musculoskeletal conditions? Please explain.

Line 229-230: This sentence does not seem to be related to the previous body of text.

Differences between the assistance dog model and AAT

Lines: 233-236: This sentence needs to be broken up:

e.g.  ‘The training and evaluation form a prescriptive model with a specific set of tasks according to the type of placement they will fulfil. These include guiding, hearing and different types of service, although many organizations also provide some level of specialty training according to the recipient's unique needs…’

Lines: 240-242: Change to: ‘Assistance dogs trained by Assistance Dogs International member organizations have standards for evaluation of the dog’s skills during its training and for the dog’s and handler’s skills after training…

Line 242: Please expand on what is meant by ‘placement-training education’.

Lines 247-250: Another painfully long and disjointed sentence. Please break it up.

Line 250: What is meant exactly by the ‘populations’ expectations of a dog? This term has been used before in the manuscript and is not immediately clear. Do you mean the number of people (clients, handlers etc.) that the dog may be exposed to?

Line 253-254: While I get your point, the sentence: ‘There are as many possible responses from dogs as there are differences in sessions’ comes across as vague and confusing.

The Moral Imperative

Lines 257-258: This sentence needs to end as:

‘…the changing culture towards the use of animals, their welfare and their well-being’

or:

‘…the changing culture towards the use of animals and their welfare and well-being

Lines 268 and 276 and 279: Why have you suddenly resorted to the first person? For line 276, it would be better to say:

‘…to decrease the risks and protect clients and the dogs that work within AAT.’

Do not use ‘we’ unless you have used it consistently throughout the paper. 

Line 282: Start the sentence beginning with: ‘Dog lifespan development…’ as a new paragraph

Also, what do you mean by the term ‘Dog lifespan development’. Would ‘Dog lifespan’ not be adequate?

Lines 282-286: This sentence is too long and needs breaking up.

Also, what do you mean by aging causing a decline in ‘general preferences’?

A Framework for Including a Dog in a Professional AAT Team

Lines 296-297: Why ‘him’? The use of pronouns becomes tricky when using ‘dog’ as a singular subject. Avoid this trap by using plural subjects i.e.: ‘The attributes of ideal dogs to work in AAT vary with what one hopes to achieve with them’.

It sounds unfair, but dogs can be referred to as ‘it’. i.e. ‘The dog removed itself from…’

Line 301: Use of ‘we’ again.

Handler profile

Lines 306-307: This is admittedly pedantic, but ‘the professional’ is singular so the pronoun should be ‘she/he’ not ‘they’. Perhaps say: ‘In AAT, it is common for professionals to own and handle the dogs they work with, and it is important to ensure they are a good match.’

Line 308: Please explain the difference between ‘welfare and well-being practices’

Lines 311-315: Even though most of it is in brackets, this sentence needs to be broken up into at least two sentences.

Job characteristics

Line 321-324: I am not sure if everything listed can be classified as ‘client population demographics’. Is the degree of immunocompromisation a demographic? Is ‘immunocompromisation’ even a word?

Line 324: Should it not be ‘the degree of immunocompromisation’? I’m also not sure how this would be determined in clients.

Lines 325-327: Section C needs rewriting or at the very least needs the punctuation to be improved. The statement: ‘…potential for unnatural human postures and equipment such as power wheelchairs…’ is particularly confusing. Readers may not comprehend what an ‘unnatural human posture’ is. Furthermore, the wording implies that a power wheelchair is also unnatural. I understand that it might appear unnatural to the dog, but I imagine that many of the components mentioned as ‘job characteristics’ would appear unnatural to the dog. Perhaps refer to power wheelchairs as ‘specialised equipmen’t?

Line 327: Put a closing bracket.

Line 328. Would a ‘psychiatric-related issue’ not fall under the ‘cognitive status’ mentioned in Section C above and determine the ‘appropriate interaction with the dog’?

Line 337: Change to ‘… working, etc.’

Line 344: Change to: ‘…a scenario plan…’?

When talking about ‘…outcomes based on different responses…to the dog…’, are you referring to responses by the handler, the client, the public? It is not clear.

Line 346: Please explain your statement; ‘…we evaluate the plausibility (which is not the same as the probability) of each scenario.’

Animal profile

Line 352: Do you mean self-motivation for pleasure? This would be a common human trait!

Line 361 and 364: Change to ‘breed-typical’

Line 373-374: Please put communicate enjoyment and self-advocacy skills as examples of consistent responses: e.g. ‘…who demonstrates consistency in its responses such as always communicating enjoyment or repeatedly removing itself from uncomfortable situations.’

Line 375-377: Please reword this sentence.

Line 377: I would say: ‘Similar to people…, rather than ‘Just like people…’

Team preparation and training

Line 393: ‘Focused’ seems to be the current preferred spelling

Lines392-395: Please break up this sentence.

Line 398: Can be reassured about what?

Line 404: The line: ‘Not every client will want to or be appropriate for AAT…’ makes no sense. Something is obviously missing.

Line 407-412: Please rewrite this whole paragraph.

Line 417: What are the ‘inservices’?

Line 418: This sentence is hard to understand. What is meant by: …’to assist with generalization of previously learned skills and default behaviors in context…’? The example given does not clear things up.

Evaluation and Re-evaluation

Line 428: Behaviour, level of interest and participation by the dog I presume?

Line 436: Change to: ‘…in which performance is found lacking.’

Line 446: Change to: ‘…to track changes and provide a useful record…’

Lines 451-453: This tortuous sentence can be rescued by breaking it into two sentences:

‘It is not recommended that participants take part in consent tests as there are safety issues for both dogs and participants. Unfamiliar handlers that are competent may, however, be used.’

Conclusions

Lines 459-450:  Change to: ‘While AAT is a structured activity with specific goals, it can also be highly varied with bespoke programs catering to the needs of the many different patients with which a single animal may work.’

Line 465: Change to: ‘…and recognize that therapy should often proceed without an animal.

Line 469: Change to: ‘…that tenet must also…’

Line 477: Change to: ‘…then annually evaluating or re-evaluating the dog.’

Author Response

Thank you for taking the time and effort in preparing such a detailed review to improve the quality of this manuscript. We have taken your comments into consideration and made changes as suggested. Responses to each of your comments are below. 

The authors evaluate the current state of animal-assisted interventions and address the lack of specific formal evaluation processes for dogs working in therapeutic settings. They also highlight the potential risks associated with not safeguarding the welfare of these working dogs and provide a useful framework for including dogs in professional AAT teams.

While the essence of this paper is sound and contains a wealth of information, I feel that it is let down by its style of writing. The text consists of numerous statements or declarations that are followed by long lists of examples in parenthesis. Along with poorly constructed sentences, this often makes for tedious and frustrating reading. While I acknowledge that the authors are clearly experts in this field, I feel that this paper would benefit by being tightened up and made a bit more interesting.  Perhaps some of the long lists that are provided in parenthesis could instead be put in tabular form. We have made grammatical revisions based on your and the other reviewers remarks. We hope you find the next iteration to be better constructed and articulated. 

Sentences are often far too long and need breaking up. The authors sometimes use the first person e.g. ‘we’ and other times not. While the use of the first person is acceptable in an article of this nature (a commentary), it is important that the authors choose a style and then remain consistent throughout. Sentences have been adjusted and all instances of first person have been removed.

Please check on Animals’ rules regarding the use of American English versus British English e.g. organization vs. organisation; specialty vs. speciality; practicing vs. practising; behavior vs. behaviour.

Please also check on the rules of in-text referencing as many of the references are cited with no date. Revised

To help improve this manuscript, I have made more specific suggestions below:

Title:

Companion Animal Welfare, Well-being and Behavior: Considerations for Selection, Evaluation and Suitability for Animal Assisted Therapy

This paper is about dogs participating in animal assisted interventions, specifically animal assisted therapy. The title should reflect this more fully. The term ‘Companion animal welfare’ is too broad. Yes, we agree. Title was changed to reflect Dog Welfare.

 Simple summary

Line 18: change to: ‘…the dogs’ welfare.’ Revised

Abstract

Line 29: By safeguarding ‘patients’, you are obviously referring to the humans participating in animal assisted therapy. In the veterinary domain, it is usual for the dog to be referred to as the patient. The way the sentence is worded, it sounds like the dogs are at risk of becoming patients if not properly matched to a job or handler. Revised 

Line 31: Couldn’t ‘disinterest in the work’ be construed as a behavioural outcome? Interesting. Yes, 'disinterest in the work' could be considered a behavioral outcome, but we felt that it was more of a 'cognitive' function than behavioral one. While viewing the sessions as potentially negative, the dog may then exhibit overt signs of stress or disinterest, but because he still gets in the car, the handler may not interpret the disinterest as problematic behaviorally. We did leave it as is, but if you feel that it needs more attention, we can accommodate.

Line 34: ‘animal assisted therapy’ can be written as AAT here Revised 

Introduction

Lines 41-53: This opening paragraph of the Introduction relies very heavily on Ernst (2014). Are there other sources of literature on the history of dog use in therapy. Is there any prior literature on the history of animal-assisted interventions in the discipline of physical disability? References added in text and in the references section

Line 45: The statement: ‘…would move away from the patient if he were tense.’ Should that not be:  he or she? Perhaps say: ‘…by lying beside non-anxious patients yet would move away from patients if they were tense.’ Revised 

Line 50: Change to: Animal Assisted Therapy (AAT) as this is the first time that the term has been used in the Introduction. Revised 

Lines 50-53: A few references would be good here (and would counter the overrepresentation of Ernst, 2014 in this paragraph). Several references were added to this section and references section

Line 63: Change to: AAT Revised 

Line 67: Change to: ‘…the constructs may apply to other species involved in AAIs.’ Revised 

Animal-Assisted Interventions

Line 72: Need a reference or two after this first sentence We felt that identifying the Google Scholar reference addressed that need. Again, if you feel it needs that change, we can accommodate.

Lines 74-78: This is an excruciatingly long sentence. Please break it into two separate sentences:

e.g.:  Though the term AAI is often the catch-all term in the literature, the environments, populations, activities, credentials of the person delivering the service, expectations of the animals, theoretical approaches and scope of sessions vary. This may complicate any prescriptive considerations for normative criteria for animal selection, preparation, evaluation, suitability and welfare in the context of the work that animals assist with. Revised 

Line 81: Change to: ‘theoretical applications’ Revised 

Lines 92 and 97 and 101: I am old fashioned and would not start a sentence with an abbreviation. I guess this is the Editor’s call.

Line 110: Change to: ‘…the United States of America…’. Revised 

Line 106-111: I would have regarded ‘Assistance animals’ or ‘service dogs’ as part of AAI. You do mention earlier that AAI is an umbrella term. I guess your point is that dogs specifically involved in tasks that mitigate the effects of an individual’s disability enjoy more legal protection than dogs involved in other forms of AAI (in the USA). Correct. Service and assistance dogs are assigned to specific individuals and trained to work with that one individual. "Therapy" dogs are trained and evaluated to work with a variety of individuals. And as you mentioned, they are afforded more legal protections than "therapy" dogs.

Line 119: Change to: AAT-specific Revised 

Lines 121-127: Another excruciating sentence! Please break it up. Revised 

Line 123: Change to ‘…found there was no consensus…’ Revised 

Line 47: I have never heard the term: ‘conflict for the best interest’ but it seems to work. Revised 

Lines 154-156: How can we be certain that the AAT guidelines have been subject to ‘independent objective verification’ and have been assessed for their scientific validity and, therefore, provide an accurate and consistent measure of how a dog works contextually in AAT? I imagine that some AAT organisations may also have a financial or reputational interest in the success of the dog, which can result in a ‘conflict for the best interest’! As you assert later: ‘there are no current, comprehensive normative evaluations for AAT due to the heterogeneous nature of intradisciplinary and interdisciplinary practices and subspecialties’. This is a good question. At this time, the competencies are in their infancy and scientific rigor is focused on that validity. Your comment about the financial interest is a legitimate concern for us as well. We hope to continue to publish and conduct research in these areas in the future, but for now, they are outside of the scope of what can be accomplished in this commentary. Hopefully this section addresses your concerns. 

Lines 169-172: Please break up this laborious sentence. Revised 

Lines 172-175: Reference(s)? Revised 

Lines 186-196: This a ten-line sentence! I think it needs to be broken up. Maybe bullet points or at the very least the use of semi colons rather than commas between each ‘expectation’. Revised 

Line 193: Insert a comma (or semi colon) between these two expectations.Revised 

Line 199: Change to: ‘…any part of the evaluation when the guardian is handling…’Revised 

Lines 202-203: Change to: ‘…dog and handler with online renewals and no re-evaluation of the initial skills required skills (ATD, 2016b; TDI, 2018), while other organizations require an in-person skills …’Revised 

Lines 217: Change to: ‘…activity, proxemics and touch, and to allow…’Revised 

Line 220: Change to: ‘… to have intermediate to advanced skills…’Revised 

Lines 218-221: This is a confusing sentence. Should it be: ‘…to have intermediate to advanced skills in AAT treatment planning and delivery and dog learning theory, training and welfare…’Revised 

or:

‘…to have intermediate to advanced skills in AAT treatment planning and delivery, dog learning theory, dog training and dog welfare…’ ??

This confusion happens fairly frequently throughout this paper. Please carefully check your placement of commas and semi colons. Revised 

Line 222-224: This is another badly phrased sentence. Perhaps something like: ‘They should be qualified to screen clients for appropriateness to participate in AAT including having the ability to determine the clients’ potential fears, allergies, health issues, cultural beliefs, history with animals and exposure to domestic violence or other traumatic events (Winkle, 2015).’Revised 

Lines 224-226: This sentence also needs tightening up.Revised 

Line 227: What does ‘people on the floor’ mean?? Unconscious people? Inebriated people? People with musculoskeletal conditions? Please explain. Revised 

Line 229-230: This sentence does not seem to be related to the previous body of text. Revised 

Differences between the assistance dog model and AAT

Lines: 233-236: This sentence needs to be broken up:

e.g.  ‘The training and evaluation form a prescriptive model with a specific set of tasks according to the type of placement they will fulfil. These include guiding, hearing and different types of service, although many organizations also provide some level of specialty training according to the recipient's unique needs…’Revised 

Lines: 240-242: Change to: ‘Assistance dogs trained by Assistance Dogs International member organizations have standards for evaluation of the dog’s skills during its training and for the dog’s and handler’s skills after training…Revised 

Line 242: Please expand on what is meant by ‘placement-training education’.Revised 

Lines 247-250: Another painfully long and disjointed sentence. Please break it up.Revised 

Line 250: What is meant exactly by the ‘populations’ expectations of a dog? This term has been used before in the manuscript and is not immediately clear. Do you mean the number of people (clients, handlers etc.) that the dog may be exposed to?Revised 

Line 253-254: While I get your point, the sentence: ‘There are as many possible responses from dogs as there are differences in sessions’ comes across as vague and confusing.Revised 

The Moral Imperative

Lines 257-258: This sentence needs to end as:Revised 

‘…the changing culture towards the use of animals, their welfare and their well-being’Revised 

or:

‘…the changing culture towards the use of animals and their welfare and well-being

Lines 268 and 276 and 279: Why have you suddenly resorted to the first person? For line 276, it would be better to say:

‘…to decrease the risks and protect clients and the dogs that work within AAT.’Revised 

Do not use ‘we’ unless you have used it consistently throughout the paper. Revised 

Line 282: Start the sentence beginning with: ‘Dog lifespan development…’ as a new paragraphRevised 

Also, what do you mean by the term ‘Dog lifespan development’. Would ‘Dog lifespan’ not be adequate?Revised 

Lines 282-286: This sentence is too long and needs breaking up.Revised 

Also, what do you mean by aging causing a decline in ‘general preferences’?Revised 

A Framework for Including a Dog in a Professional AAT Team

Lines 296-297: Why ‘him’? The use of pronouns becomes tricky when using ‘dog’ as a singular subject. Avoid this trap by using plural subjects i.e.: ‘The attributes of ideal dogs to work in AAT vary with what one hopes to achieve with them’.

It sounds unfair, but dogs can be referred to as ‘it’. i.e. ‘The dog removed itself from…’ Yes, this is true; however, we feel that referring to dogs as "it" depersonalizes them and negates them as partners as well as the argument that their welfare needs and concerns need to be considered. 

Line 301: Use of ‘we’ again. Revised 

Handler profile

Lines 306-307: This is admittedly pedantic, but ‘the professional’ is singular so the pronoun should be ‘she/he’ not ‘they’. Perhaps say: ‘In AAT, it is common for professionals to own and handle the dogs they work with, and it is important to ensure they are a good match.’ Revised 

Line 308: Please explain the difference between ‘welfare and well-being practices’ Revised 

Lines 311-315: Even though most of it is in brackets, this sentence needs to be broken up into at least two sentences. Revised 

Job characteristics

Line 321-324: I am not sure if everything listed can be classified as ‘client population demographics’. Is the degree of immunocompromisation a demographic? Is ‘immunocompromisation’ even a word? Revised 

Line 324: Should it not be ‘the degree of immunocompromisation’? I’m also not sure how this would be determined in clients. Revised 

Lines 325-327: Section C needs rewriting or at the very least needs the punctuation to be improved. The statement: ‘…potential for unnatural human postures and equipment such as power wheelchairs…’ is particularly confusing. Readers may not comprehend what an ‘unnatural human posture’ is. Furthermore, the wording implies that a power wheelchair is also unnatural. I understand that it might appear unnatural to the dog, but I imagine that many of the components mentioned as ‘job characteristics’ would appear unnatural to the dog. Perhaps refer to power wheelchairs as ‘specialised equipmen’t? Revised 

Line 327: Put a closing bracket. Revised 

Line 328. Would a ‘psychiatric-related issue’ not fall under the ‘cognitive status’ mentioned in Section C above and determine the ‘appropriate interaction with the dog’? Revised 

Line 337: Change to ‘… working, etc.’Revised 

Line 344: Change to: ‘…a scenario plan…’?Revised 

When talking about ‘…outcomes based on different responses…to the dog…’, are you referring to responses by the handler, the client, the public? It is not clear.Revised 

Line 346: Please explain your statement; ‘…we evaluate the plausibility (which is not the same as the probability) of each scenario.’Revised 

Animal profile

Line 352: Do you mean self-motivation for pleasure? This would be a common human trait! Revised 

Line 361 and 364: Change to ‘breed-typical’Revised 

Line 373-374: Please put communicate enjoyment and self-advocacy skills as examples of consistent responses: e.g. ‘…who demonstrates consistency in its responses such as always communicating enjoyment or repeatedly removing itself from uncomfortable situations.’Revised 

Line 375-377: Please reword this sentence.Revised 

Line 377: I would say: ‘Similar to people…, rather than ‘Just like people…’Revised 

Team preparation and training

Line 393: ‘Focused’ seems to be the current preferred spellingRevised 

Lines392-395: Please break up this sentence.Revised 

Line 398: Can be reassured about what?Revised 

Line 404: The line: ‘Not every client will want to or be appropriate for AAT…’ makes no sense. Something is obviously missing.Revised 

Line 407-412: Please rewrite this whole paragraph.Revised 

Line 417: What are the ‘inservices’?Revised 

Line 418: This sentence is hard to understand. What is meant by: …’to assist with generalization of previously learned skills and default behaviors in context…’? The example given does not clear things up.Revised 

Evaluation and Re-evaluation

Line 428: Behaviour, level of interest and participation by the dog I presume?Revised 

Line 436: Change to: ‘…in which performance is found lacking.’Revised 

Line 446: Change to: ‘…to track changes and provide a useful record…’Revised 

Lines 451-453: This tortuous sentence can be rescued by breaking it into two sentences:Revised 

‘It is not recommended that participants take part in consent tests as there are safety issues for both dogs and participants. Unfamiliar handlers that are competent may, however, be used.’Revised 

Conclusions

Lines 459-450:  Change to: ‘While AAT is a structured activity with specific goals, it can also be highly varied with bespoke programs catering to the needs of the many different patients with which a single animal may work.’Revised 

Line 465: Change to: ‘…and recognize that therapy should often proceed without an animal.Revised 

Line 469: Change to: ‘…that tenet must also…’Revised 

Line 477: Change to: ‘…then annually evaluating or re-evaluating the dog.’Revised 

Round 2

Reviewer 3 Report

Second review

Dog Welfare, Well-being and Behavior: Considerations for Selection, Evaluation and Suitability for Animal Assisted Therapy

General comments

These revisions have improved the manuscript slightly, although I found that it still consists of long and obtuse sentences. Often, I was left confused and had to read and re-read the sentence before understanding what the authors meant. This manuscript would, therefore, benefit further from being tightened up and made less “flabby”. An example of this falls between lines 307-311. The first part of the paragraph: ‘The lifespan of the dog is a frequently overlooked area. Dogs live such a relatively short life compared to humans. They age faster so there is the potential to see a decline in physical skills and abilities, cognitive and emotional processes, and general preferences of likes and dislikes. In other words, as dogs age and experience declining vision or hearing and decreased mobility issues, their tolerances and preferences change (Barker et al, 2019).’

Could be tightened up into:

‘The shorter lifespan of the dog, compared to humans, remains a frequently overlooked area. Faster aging can result in a notable decline in the dog’s physical skills and abilities, cognitive and emotional processes, and general preferences of likes and dislikes. Furthermore, as dogs age and experience declining vision, hearing and mobility, their tolerances and preferences may change (Barker et al, 2019).’  

There are many examples throughout the manuscript of sentences written in this long and obtuse fashion.

The in-text citations need to be consistent. Sometimes all the authors are provided and other times only the first author (with an et al. afterwards) is given. Two authors are sometimes separated with an ‘and’ and other times with an ‘&’. This may be an academic point as in this journal the in-text referencing is provided as numbered superscripts.

Check for use of a hyphen in ‘AAT-specific’ throughout

More specific changes are recommended below:

Lines 44-46: I think that this sentence would read better as: ‘…where Freud found that Jofi helped facilitate sessions by lying beside non-anxious patients and moving away from tense or stressed patients (Ernst, 2014; Pellegrini, 2009).’

Lines 52-57: I’m sure some of these references can be shortened to first author followed by et al. e.g. Foreman, Allison, Poland, Jean Meade, and Wirth, 2019 could be referred to as Foreman et al. 2019

Line 72: Close brackets after Wimer 2018

Line 105:  Cite your reference as Kerulo et al.

Lines: 106-108: Change to: ‘…for engaging with their clients or student, such as obtaining informed consent, offering formal evaluation, establishing short- and long-term goals, and measuring and documenting progress.’

Line114: Change to: ‘…considered formal therapy dogs (Wlodarczyk, 2019).’

Line127: Change to: ‘…makes it difficult to identify AAT-specific processes and literature.’

Line 129: Change to: AAT-specific

Line 141: I would start: ‘It is fair to say that AAA, AAT and…’ as a new paragraph

Line 159: Remove the semi-colon and make: ‘Related protocols may also not have been assessed for their scientific validity (Mongillo et al., 2015).’ a separate sentence.

Line 161: Change to: an AAT capacity

Lines 166-167: Several professional organizations have made progress in publishing AAT-specific competencies while others have included AAT as a recognized practice area.

Line 170: Change to: ‘…recently released a list of required competencies and ethical guidelines…’

Lines183-184: This sentence makes no sense: ‘The messages about the dogs’ wants and needs being important that are conveyed to the clients are often at the crux of effective treatment.’ Please modify

Line 199: Change to: ‘…the following activities:’

Lines 200-215: Make each bullet point start with a capital letter (as they are not the continuation of a sentence). Remove ‘to’ at the beginning of each bullet point

e.g. Line 202: Change to

  • ‘Avoid vocalizations such as …’

Line 218-219: Change to: ‘Similarly, demonstrations of pulling, shyness, or resisting any part of the evaluation when the guardian is handling the dog would also mean automatic failure.

Line 223: Change to: ‘role play-based’

Line 225: Change to: ‘…measured the difference between evaluated and registered AAI dogs…’

Lines 240-241: What is meant by ‘treatment planning and delivery’?

Line 251: Why have you put AAT in brackets after ‘paid working roles’?

Also change to: ‘…who incorporate their dogs into practice…’

Line 267-268: Change to: ‘Dogs may be expected to independently seek out and greet clients and even include barking or other vocalizations in the greeting.’

Line 268-270: Sorry I have no idea what this sentence means. Please reword

Line 271-272: A terrible sentence: Perhaps instead say: ‘While they are expected to have manners and obedience, ideally the dogs should be able to choose whether or not they participate.’

Perhaps also explain what they would (or would not) participate in.

Line 338: Remove bracket

Lines 364-365: I do not see the relationship between dogs’ requirements for having a space away from humans and activity for rest AND dogs enjoying routines. Why are these two separate facts separated by a semi-colon?

Line 365: Are the activities in brackets examples of ‘participation expectations’? if so, put e.g. at the beginning

Also replace the hyphen with a comma

Lines 368-372: This sentence is too long and confusing at the end. Please reword.

Lines 398-401: Another dreadful sentence. A trait is the enjoyment of learning and training tasks and demonstrating consistency in responses. The sentence has been worded to indicate that the trait is the dog enjoying the learning and training tasks and the dog demonstrating consistency in responses. A dog cannot be a trait. Please reword.

Lines 402-404: This sentence is flabby and needs to be broken up and reworded: e.g. ‘A dog will typically need to possess a sociability towards unfamiliar individuals and a curiosity towards various activities. The dog should also demonstrate patience towards participants who fail to complete an activity that would normally bring gratification to the dog.’

Lines 412: Change to: ‘…large strides can be made towards achieving this.’

Line 413: Return again and again to what?

Line 422-424: Please reword this sentence

Line 429: Change to: ‘…the ability to advocate for the dog when necessary.’

Line 432: Change to: ‘…things the dog enjoyed at 2 years old may be very different at 7 years old.

Line 440: Change to: ‘…to engage in an activity that involves…’

Line 449: What skills are you referring to when you say: ‘both previously learned skills’?

Line 450-451: Put your example in brackets

Line 490: Change to: ‘… it can also be highly varied with bespoke…’

Line 500: put Do No Harm in inverted commas

Line 506: Does ‘her’ refer to the professional or the dog? Either way, it would be better to use ‘their’

Line 509: Change to: ‘- ideally on an annual basis.’

Author Response

Thank you for your thorough review. The changes have been made to the manuscript and uploaded. It is our hope that this version is suitable. Responses to your suggestions are below in orange. Thank you.

Second review

Dog Welfare, Well-being and Behavior: Considerations for Selection, Evaluation and Suitability for Animal Assisted Therapy

General comments

These revisions have improved the manuscript slightly, although I found that it still consists of long and obtuse sentences. Often, I was left confused and had to read and re-read the sentence before understanding what the authors meant. This manuscript would, therefore, benefit further from being tightened up and made less “flabby”. An example of this falls between lines 307-311. The first part of the paragraph: ‘The lifespan of the dog is a frequently overlooked area. Dogs live such a relatively short life compared to humans. They age faster so there is the potential to see a decline in physical skills and abilities, cognitive and emotional processes, and general preferences of likes and dislikes. In other words, as dogs age and experience declining vision or hearing and decreased mobility issues, their tolerances and preferences change (Barker et al, 2019).’

Could be tightened up into:

‘The shorter lifespan of the dog, compared to humans, remains a frequently overlooked area. Faster aging can result in a notable decline in the dog’s physical skills and abilities, cognitive and emotional processes, and general preferences of likes and dislikes. Furthermore, as dogs age and experience declining vision, hearing and mobility, their tolerances and preferences may change (Barker et al, 2019).’   This has been modified and sentences that were unclear or too long have been amended.  

There are many examples throughout the manuscript of sentences written in this long and obtuse fashion.

The in-text citations need to be consistent. Sometimes all the authors are provided and other times only the first author (with an et al. afterwards) is given. Two authors are sometimes separated with an ‘and’ and other times with an ‘&’. This may be an academic point as in this journal the in-text referencing is provided as numbered superscripts. The references have been adjusted in-text. All references with two authors have both names included with the ampersand. References with more than two authors have been changed to first last name et al, year. We hope this is sufficient.   

Check for use of a hyphen in ‘AAT-specific’ throughout
This has been modified. We did a search and it seems that all references have been addressed.

More specific changes are recommended below:

Lines 44-46: I think that this sentence would read better as: ‘…where Freud found that Jofi helped facilitate sessions by lying beside non-anxious patients and moving away from tense or stressed patients (Ernst, 2014; Pellegrini, 2009).’ Revised

Lines 52-57: I’m sure some of these references can be shortened to first author followed by et al. e.g. Foreman, Allison, Poland, Jean Meade, and Wirth, 2019 could be referred to as Foreman et al. 2019 All have been changed to reference first author et al, year. 

Line 72: Close brackets after Wimer 2018 Revised 

Line 105:  Cite your reference as Kerulo et al. Revised 

Lines: 106-108: Change to: ‘…for engaging with their clients or student, such as obtaining informed consent, offering formal evaluation, establishing short- and long-term goals, and measuring and documenting progress.’ Revised 

Line114: Change to: ‘…considered formal therapy dogs (Wlodarczyk, 2019).’ Revised 

Line127: Change to: ‘…makes it difficult to identify AAT-specific processes and literature.’ Revised 

Line 129: Change to: AAT-specific Revised 

Line 141: I would start: ‘It is fair to say that AAA, AAT and…’ as a new paragraph Revised 

Line 159: Remove the semi-colon and make: ‘Related protocols may also not have been assessed for their scientific validity (Mongillo et al., 2015).’ a separate sentence. Revised 

Line 161: Change to: an AAT capacity Revised 

Lines 166-167: Several professional organizations have made progress in publishing AAT-specific competencies while others have included AAT as a recognized practice area. Revised 

Line 170: Change to: ‘…recently released a list of required competencies and ethical guidelines…’ Revised 

Lines183-184: This sentence makes no sense: ‘The messages about the dogs’ wants and needs being important that are conveyed to the clients are often at the crux of effective treatment.’ Please modify This sentence has been changed to this to clarify...
When an AAI clinician honors and acts upon the messages shared by their dogs about the dogs’ wants and needs, their clients witness the compassion and care extended to the dogs to which clients can generalize to themselves. The safety that clients feel with the clinicians is often at the crux of effective treatment. Conversely, ignoring the needs of the dogs can send potentially harmful messages to clients about their own needs, wants and self-advocacy. 

Line 199: Change to: ‘…the following activities:’ Revised

Lines 200-215: Make each bullet point start with a capital letter (as they are not the continuation of a sentence). Remove ‘to’ at the beginning of each bullet point Revised

e.g. Line 202: Change to

  • ‘Avoid vocalizations such as …’

Line 218-219: Change to: ‘Similarly, demonstrations of pulling, shyness, or resisting any part of the evaluation when the guardian is handling the dog would also mean automatic failure.Revised

Line 223: Change to: ‘role play-based’ Revised

Line 225: Change to: ‘…measured the difference between evaluated and registered AAI dogs…’ Revised

Lines 240-241: What is meant by ‘treatment planning and delivery’? Revised

Line 251: Why have you put AAT in brackets after ‘paid working roles’?

Changed to:

Furthermore, some AAA organization policies are very clear that the AAA-related insurance policies may not cover handlers and dogs who participate in animal assisted therapy in paid working roles such as mental health providers who incorporate her dogs into practice (ATD, 2016a; Pet Partners, n.d.-c; TDI, 2017).

Also change to: ‘…who incorporate their dogs into practice…’ Revised

Line 267-268: Change to: ‘Dogs may be expected to independently seek out and greet clients and even include barking or other vocalizations in the greeting.’ Revised

Line 268-270: Sorry I have no idea what this sentence means. Please reword

Changed to:
Dogs may be expected to independently seek out and greet clients and even include barking or other vocalizations in the greeting. For example, clients who do not typically receive enthusiastic reactions from others might feel very special having a dog bark excitedly to them when they enter the room. The verbal and physical excitement may strengthen the initiation and facilitation of the therapeutic process

Line 271-272: A terrible sentence: Perhaps instead say: ‘While they are expected to have manners and obedience, ideally the dogs should be able to choose whether or not they participate.’ Revised

Perhaps also explain what they would (or would not) participate in. Revised, added "...choose whether or not they participate in the AAI session."

Line 338: Remove bracket Revised

Lines 364-365: I do not see the relationship between dogs’ requirements for having a space away from humans and activity for rest AND dogs enjoying routines. Why are these two separate facts separated by a semi-colon? Revised

Line 365: Are the activities in brackets examples of ‘participation expectations’? if so, put e.g. at the beginning Revised

Also replace the hyphen with a comma Revised

Lines 368-372: This sentence is too long and confusing at the end. Please reword.

Lines 398-401: Another dreadful sentence. A trait is the enjoyment of learning and training tasks and demonstrating consistency in responses. The sentence has been worded to indicate that the trait is the dog enjoying the learning and training tasks and the dog demonstrating consistency in responses. A dog cannot be a trait. Please reword.

Changed to:
Participation expectations regarding the types of activities in which dogs will be participating (e.g. talk therapy, dog training, physical activities between dogs and humans) also need to be clearly articulated. In some situations, there may be more than one dog available to participate in sessions. In these settings, each dog would require an individual profile with the dog’s needs and preferences clearly identified to determine which dog would be best for each interaction. Each therapy session should also have a session plan (Shoemaker, 1995) with desired outcomes of that session that factor in the potentially different responses to the dog by the client and by the dog to the client where each specific intervention proposed to evaluate plausibility. 

Lines 402-404: This sentence is flabby and needs to be broken up and reworded: e.g. ‘A dog will typically need to possess a sociability towards unfamiliar individuals and a curiosity towards various activities. The dog should also demonstrate patience towards participants who fail to complete an activity that would normally bring gratification to the dog.’ Revised

Lines 412: Change to: ‘…large strides can be made towards achieving this.’  Revised

Line 413: Return again and again to what? Revised. Changed to:

If practitioners are doing AAT correctly, the dog will display signs of enjoyment and want to return to sessions to work with participants again and again.  

Line 422-424: Please reword this sentence

Changed to:

Handlers should use humane, positive reinforcement training techniques that do not involve force or coercion as this will build a dog's confidence for the complex nature of AAT. The handler is seen as a secure base for the dog from which to operate (Mariti et al., 2013) and using training techniques that harm or frighten the dog can damage that secure base.

Line 429: Change to: ‘…the ability to advocate for the dog when necessary.’ Revised

Line 432: Change to: ‘…things the dog enjoyed at 2 years old may be very different at 7 years old. Revised 

Line 440: Change to: ‘…to engage in an activity that involves…’ Revised 

Line 449: What skills are you referring to when you say: ‘both previously learned skills’? Revised 

Changed to:
AAT session simulations are a great way to assist with generalization of previously learned skills or the default behaviors required in a given context (e.g. if a dog sees a yoga mat being unrolled, this may signal that it should go and lay at the end of it, wherever this action occurs). 

Line 450-451: Put your example in brackets Revised

Line 490: Change to: ‘… it can also be highly varied with bespoke…’ Revised 

Line 500: put Do No Harm in inverted commas Revised 

Line 506: Does ‘her’ refer to the professional or the dog? Either way, it would be better to use ‘their’ Revised 

Line 509: Change to: ‘- ideally on an annual basis.’ Revised 
